# 1000 Days: The “WeCare Generation” Program—The Ultimate Model for Improving Human Mental Health and Economics: The Study Protocol

**DOI:** 10.3390/ijerph192416741

**Published:** 2022-12-13

**Authors:** Orlando Uccellini, Andrea Benlodi, Emanuele Caroppo, Loredana Cena, Gianluca Esposito, Isabel Fernandez, Maria Ghazanfar, Antonio Imbasciati, Francesco Longo, Marianna Mazza, Giuseppe Marano, Renata Nacinovich, Antonio Pignatto, Arthur Rolnick, Marco Trivelli, Elena Spada, Cinzia Vanzini

**Affiliations:** 1Department of Mental Health, ASST Brianza, 20871 Vimercate, Italy; 2Clinical Psychology Unit Carlo Poma Hospital, ASST Mantova, 46100 Mantua, Italy; 3Department of Mental Health, Local Health Authority Roma 2, 00159 Rome, Italy; 4Department of Clinical and Experimental Sciences, University of Brescia, 25123 Brescia, Italy; 5Department of Psychology and Cognitive Science, University of Trento, 38122 Trent, Italy; 6EMDR Italia Association, 20814 Varedo, Italy; 7Maternal and Child Department, ASST Brianza, 20871 Vimercate, Italy; 8Cergas Center for Research on Health and Social Care Management, SDA Bocconi University, 20136 Milan, Italy; 9Department of Geriatrics, Neuroscience and Orthopedics, Institute of Psychiatry and Psychology, Fondazione Policlinico Universitario A. Gemelli IRCCS, Universita’ Cattolica del Sacro Cuore, 00168 Rome, Italy; 10Child and Adolescent Neuropsichiatry, ASST Monza, NeuroMI—Milan Center for Neuroscience, University of Milano Bicocca, 20126 Milan, Italy; 11Department of Psychology, IUSTO—Salesian University Institute Torino Rebaudengo, 10155 Turin, Italy; 12Department of Economics, Humphrey School of Public Affairs, University of Minnesota, Minneapolis, MN 55455, USA; 13General and Economic Direction, ASST Brianza, 20871 Vimercate, Italy; 14Magistica, 23875 Lecco, Italy; 15Training Sector Management, ASST Brianza, 20871 Vimercate, Italy

**Keywords:** adverse childhood experiences, prevention, trauma, economic, welfare, child, gross domestic product, pregnancy, parental trauma

## Abstract

Introduction: The COVID-19 pandemic stressed the necessity of a new resilience of the human population and health system. The “WeCare Generation” program is a new proposal of territorial intervention, with a new paradigm, on the diseases of the human body and mind. Background: In recent decades, the independent strands of investigation on brain plasticity and early trauma consequences have demonstrated that traumatic experiences in the period from pregnancy to the age of 3 years have an enormous impact on an individual’s future development, and both physical and mental health. Research shows that adverse child experiences (ACEs) are associated with a strong risk of conditions such as: harmful alcohol use, smoking, illicit drug use, high body-mass index, depression, anxiety, interpersonal violence, cancer, type 2 diabetes, cardiovascular diseases, stroke respiratory diseases and, as a consequence, to a high financial cost in Italy and also across Europe (1–9% GDP) and the USA (total annual costs estimated to be USD 581 billion in Europe and USD 748 billion in North America). All this suggests that an early intervention on that traumatized-slice of population leads to multiplied savings. Methods: A multi-center, randomized, controlled trial was designed. The parents of the future neonatal population (from pregnancy to delivery) with trauma will be enrolled, and randomized to treatment, or control arm. The article describes in detail how the primary outpoint (cost to the national health system), and some secondary outpoints, will be collected. Discussion: An overall rate of return on investment (ROI) statistically significant 13.0% per annum with an associated benefit/cost ratio (BCR) of 6.3 is expected as the primary outcome of the “WeCare Generation” program. Our proposed model predicts a new medical paradigm aiming to empower new generations, with a strong return on economy and health.

## 1. Introduction

In the past two years, the COVID-19 pandemic had a strong impact not only because of the effect of the virus on the human organism, but also as a global traumatic event. Significant changes in lifestyles with deleterious consequences on physical and mental health, the development of anxiety and depressive symptoms in the general population, deterioration in life satisfaction, the considerable aggravation of the psychic and physical conditions of patients already suffering from mental disorders, and widened health inequalities and inequities, represent just a small portion of the outcomes of this worldwide event [1,2,3,4,5,6,7,8,9].

Trauma represents for the individual a psychically unaddressed situation that elicits and unconsciously ‘repeats’ past traumas [10]. A decisive factor, therefore, for individual resilience, is how well past traumas, especially those involving childhood, have been processed. On the other hand, a reliable and caring environment during childhood is a basic element of resilience [11].

As a global tragic event, the COVID-19 pandemic has posed a serious threat to global mental health [8,9]. The current evidence and published literature related to previous epidemics suggest that mental health issues may arise after the peak of the pandemic [6], with increased prevalence among the vulnerable population and people with risk factors [3]. Currently we observe an exponential increase in mental illnesses especially in the adolescents, but also of the adult and early childhood ages [8,9,10]. Growing demands for mental health, urgent social protection, and better public health systems have posed critical challenges in both economic and service delivery capacities [4,5,6].

Starting from scientific evidence, it is possible to hypothesize an area-based preventive-curative model that could lead to a significant decrease in the risk of developing major mental and physical disorders and simultaneously could guarantee economic savings. In such perspective, the designed program should favor an enhanced resilience and productivity of the population with an inter-generational effect. The “WeCare Generation” is based on a new proposal of territorial intervention, with a novel paradigm of prevention and care of the diseases affecting the human body and mind. It aims to increase human potential in both health and productivity in the most plastic period of a whole life cycle, intervening on early trauma, and empowering new generations, with a strong economic and health return.

The Program proposes a workable model to detect and treat trauma on parental couples with their newborn until their 3rd year of life in order to improve physical and mental health with a high rate of welfare and socio-economic return.

In today’s routine standard care, there is no pathway for the early diagnosing and treatment of trauma. Usually, the effects on the newborn emerge during later stages of development, often in adolescence or adulthood. In addition, trauma is not adequately considered as an important risk factor for numerous physical diseases.

The program consists of a psychological treatment protocol for traumatic events through the use of Eye Movement Desensitization and Reprocessing (EMDR). In addition, a schedule of repeated home visits by parental coaches is planned. EMDR is an integrative psychotherapy approach that has been extensively researched and proven effective for the treatment of traumatic events and its consequences, such as Post-Traumatic Stress Disorder (PTSD) [12]. It consists of a therapy that focuses on memories of traumatic and/or stressful experiences contributing to mental disorders or psychological problems and enables people to heal from the symptoms and emotional distress that are the result of disturbing life experiences.

### Background

Adverse childhood experiences (ACEs) are potentially traumatic events that occur in childhood (0–17 years) [10]. ACEs include physical abuse, sexual abuse, psychological abuse, physical neglect, psychological neglect, witnessing domestic abuse, having a close family member who misused drugs or alcohol, having a close family member with mental health problems, having a close family member who served time in prison, and parental separation or divorce on account of relationship breakdown. Toxic stress from ACEs can change brain development and affect how the body responds to stress. ACEs are linked to chronic health problems, mental illness, and substance misuse in adulthood. However, ACEs can be prevented [10].

The Adverse Childhood Experiences study [11] and the subsequent body of research provide compelling evidence that the risk of adverse health consequences increases as a function of the number of categories of adversities adults were exposed to in childhood.

ACEs have in Europe and in Italy a prevalence of about 20% [13]. Traumatized individuals and their children require an amount of resource from the welfare and health system. ACEs are an important risk factor for various mental, physical, and social [14,15,16,17] conditions in infancy and adulthood and they have a significant impact with a high financial cost in Italy and also across Europe (1–9% GDP) and the USA (total annual costs estimated to be USD 581 billion in Europe and USD 748 billion in North America) [18,19].

Furthermore, there is independent research showing the effectiveness to invest in the period from pregnancy up to the age of 3 of the child, for a very high return (13–18%) [20]. This is in fact the period of major human brain plasticity, in which the bound with the caregiver (and the trauma) has a huge imprinting on the epigenetics [21,22,23] and neurological-behavioral organization of the child [24,25,26].

Parental functioning and attachment are in fact impaired by ACEs [27] and there are many studies focusing on the rebound of ACEs on the subsequent newborn generation [28,29,30].

Finally, some follow up studies show that many health and social conditions (also associated with ACEs) are clustered and a slice of about 20% of the population seems to “consume” 80% of welfare resources [31,32]. All this suggests that an early intervention on that traumatized portion of population leads to multiplied savings.

However, there is a lack of reliable follow-up studies on how much interventions on mothers/parents (improving interaction and parental care) can affect the development of children in social and economic terms [33,34]. Furthermore, there is little consideration on the treatment of parents’ internal emotive distorted relationships due to their own traumatic experiences during childhood. It is known that parental care functioning is a co-regulator of infant physiological and emotional homeostasis Understanding parental regulation of the infant’s immediate neurobehavioral functioning within the context of attachment quality, that may provide insights into the complex processes during early life, initiating the pathway to pathology [35].

We propose a workable model to detect and treat ACEs on parental couples with their newborn until their 3rd year of life, that could improve physical and mental health of the population, with a potential high rate of health and socioeconomic return.

In particular, the trauma treatment is designed to heal the internal attachment relationship within parents [36] and within the child, often distorted by traumatic experiences. The hypothesis is that this treatment could positively affects the quality of parental care.

## 2. Materials and Methods

The study has been designed as a multi-center randomized no-mask controlled trial and is at the same time an economic and health research. It will involve a total of 19 obstetrical recruitment centers in Lombardy in the province of Monza Brianza (16 territorial and 3 hospital points) and will be headed by a central coordinating group located in Carate Brianza (led by three professionals: O.U., C.V., and M.G.).

### 2.1. Inclusion Criteria

All families with an Adverse Childhood Experience before or during pregnancy (delivery included) will be consecutively enrolled in the study. Single parents will also be included.

### 2.2. Exclusion Criteria

Families with major language barriers, adopted children, or in which a member participates in other studies will be excluded. Neonates with fetal hydrops and major congenital anomalies diagnosed during pregnancy or established at the time of delivery, fetal hydrops, and major congenital anomalies of one sibling will be also excluded.

### 2.3. Recruitment

Usually, in the public health care system in Lombardy, a pregnant woman, together with the father-to-be, in the first trimester of pregnancy, requests the first meeting with the midwife to perform an initial anamnestic collection and measure the first parameters. This is the gateway to the ‘birth pathway’ already routine in Italy.

The program will be proposed to all parents accessing ASST Brianza birth points in the three years 2023–2026 through the first Obstetrical Booking meeting, at the time of booking.

The midwife will inform the couple that a study is being conducted on prevention in the first three years on the effect of parental and perinatal trauma and will propose consent to the study. Sufficient time will be allowed for consent.

If the couple adheres to the study, socioeconomic data will be collected through an instrument already used in previous studies approved by the Italian National Institute of Health [37]. In this scale the economic condition is ranked from 1 to 4 as follows: (1) serious problem (debts, cannot pay rent, etc.); (2) some problems (limitation of daily expenses, cannot afford vacations); (3) more modest standards, but no particular difficulties; and (4) medium to high (home ownership, vacation, etc.). A psychologist will collect the traumatic aspects by applying appropriate instruments together with the midwife.

### 2.4. Trauma Detection

During the interview with the psychologist, the segment of the population of traumatized parents will be surveyed. Trauma screening will be done with the ACE questionnaire (standardized for adverse childhood experiences) and the Trauma History Questionnaire (through a semi-structured interview by the psychologist).

A score ≥ 2 on the ACE questionnaire or the presence of even one trauma factor on the Trauma History Questionnaire will lead to those parents being considered as ‘traumatized’ in our study.

The same semi-structured interview will be repeated later at 28 weeks’ gestation and at 10 days of the infant’s life to make sure to detect other potential adverse events that have occurred.

It will be explained to enrolled subjects that if trauma is present or was detected they will be included in the study and randomized into the control or treatment arm.

### 2.5. Randomization

Eligible mothers/couples will be allocated to one of the two arms (treated or experimental, and control) by block randomization. A software has been designed to automatically generate a randomization code and to obtain, at each birth point, a balance between subjects with premature births or labor trauma and age of mothers (≤16 years, 17–39 years, ≥40 years). An ad-hoc randomization software will be available for each birth point, on a password-protected specific website. The randomized code sequence will be blinded for all midwife, psychologist and anyone involved in the study. The randomization of each mother/parent will take place as soon the ACEs will be detected: at initial meeting if trauma has already occurred, or at the first visit after ACE.

### 2.6. Sample Size

In the 19 obstetrical recruitment centers involved in this study, the number of deliveries is about 3500, of which about 80% (2800) have no exclusion criteria, every year. Of these 2800/year, about 20% (560) will have ACE.

A recruitment time of 3 years is suggested and about 1680 mothers/parents will be expected to be included in the study (840 per arm, Intention to Treat population).

Assuming a loss to follow-up (drop-out) of about 40% between pregnancy and three 3 years of the child’s life, complete data for about 1008 mothers/parents will be expected (per protocol population)

### 2.7. Monitoring and Data Collection

All the data will be collected using the ad-hoc website. At the end of data collection, the data will be downloaded in a single dataset for the analysis. The statistician will be blinded. Data of all pregnancies will be collected from initial meeting with the obstetrician to birth. The data will be collected to 3 years of age if the mother/parents will be included in the study. Figure 1 shows the study design. Each birth point will adopt its protocol for pregnancy management with the exception of the psychologist interviews.

### 2.8. Ethical Considerations

Standard care, defined by the regional health system, will be provided to all infants and families, regardless of whether or not they participate in the study. This study could demonstrate a protective effect of early intervention on families with trauma, which consequently also leads to savings in health care spending. If so, it could be the impetus for redefining standard care in the regional health care system. Enrolled subjects will sign informed written consent. The whole process will be conducted in accordance with the Declaration of Helsinki. The research protocol has been approved by the local Brianza Ethics Committee (Monza, Italy). The Ethics Committee is an independent body responsible for ensuring the protection of the rights, safety, and well-being of trial subjects and for providing public assurance of that protection. The anonymity of study participants will be ensured through an algorithm of assigning a code to each individual.

### 2.9. Intervention

#### 2.9.1. Control Arm

The control group will follow the ‘routine birth pathway’, which already exists in the Lombardy region and is developed from the beginning of pregnancy until about 6 to 8 weeks after delivery, and is coded and different according to the risk of the pregnancy. Survey of the variables of interest will be done through the tax code and telephone interviews.

#### 2.9.2. Experimental Arm

Additional treatment from pregnancy up to 3 years of the child will be added to the basic treatment that controls also receive.

The “treatment protocol” for traumatized parents and their children will be common for all the obstetrical recruitment centers. It consists of 3 parallel lines of support, which are home visiting, parent groups, and focal treatment of the traumatic experience.

The home visiting, conducted by the parenting coach, will last 1 h, and will have a scheduled frequency depending on the period. The purpose of the visits will be mainly to strengthen the parent-child relationship. The new figure of the parenting coach will be a kind of “aunt or uncle” or “grandmother or grandfather” and will try from the beginning to establish a bond of trust with the new parents. This professional figure will accompany at the territorial domestic level the couple of parents in the period extending from pregnancy until the third year of life of the newborn.

A protocol of training and monthly psychological supervision for this figure has been prepared. The protocol explains also in detail the support activities such as listening, psychoeducation, coordination, and the accompaniment of the couple at various stages from pregnancy to the third year of life. The parent groups will be carried out every three months from pregnancy to the first 3 years of the child’s life for a total of 15 sessions. They will be held by a psychologist experienced in group therapy with the aim to promote the couples’ internal resources and to process the individuals’ internal relationships with their own father and mother. Last, but not least, the focal treatment of trauma will be realized through 20 individual Eye Movement Desensitization and Reprocessing (EMDR) sessions on the unprocessed traumatic target. The treatment will be will be chosen, on the basis of common protocols, depending on the type of trauma and the needs of the mother and father.

### 2.10. Training

To standardize data collection and treatment among the various obstetric and birth points scattered throughout the territory, we drew up a selection, based on specific characteristics of the providers involved in the study. For the latter, we also designed training and used guiding protocols.

EMDR therapy is a structured evidence based therapy that encourages the patient to briefly focus on the trauma memory while simultaneously experiencing bilateral stimulation (typically eye movements), which is associated with a reduction in the vividness and emotion associated with the trauma memories.

EMDR treatment itself is already standardized through protocols and various stages of trauma processing and allows for its uniform execution among different therapists. As for group meetings, there will be a protocol that specifies for each meeting the procedure to be conducted, the psychoeducational goals, and the targets to be addressed [38].

The psychologists involved in the study (those involved in the administration of the ACE questionnaire and the trauma history questionnaire or those who will treat the parents) must be, at a basic level, already accredited in psychotraumatology in the EMDR association (first and second level) and must have at least 2 years of working experience in psychotraumatology.

Psychologists who will be in charge of group meetings must additionally have certification for group EMDR therapy from the association.

The parental coach will be subjected to some tests (ACE score, Adult Attachment Interview) to measure his attachment patterns and the absence of important unprocessed trauma cores. A formal assent of the parental coach to be supervised by a psychologist about his internal feelings that might arise during his home visiting intervention is equally important. The parental coach will conduct an eight-meeting training aimed at conveying the methods of home visiting, codified by a special protocol.

### 2.11. Primary and Secondary Endpoints

The purpose of our study is to estimate the association between early intervention on parental trauma and possible savings in health care costs (primary endpoint).

There are also some important secondary social-healthcare endpoints. Several birth points from a large geographical area of the Lombardy region (Brianza) coordinated by a central core will be involved in the study. Treatment of parental trauma will be in addition to the routine ‘birth pathway’ that is, a set of routine interventions that are already in place and scheduled in Lombardy. The group of parents supplemented with trauma treatment will be compared with the untreated control group performing only the routine intervention.

Treatment and data collection for the primary and secondary outcomes of the study will be harmoniously grafted into the interventions already scheduled in the Italian Health System (obstetric bookings and controls, pediatric health control, vaccination appointments).

With an early-trauma base intervention we expected a short- and long-term impact on the life cycle and on the next generation newborn [39,40,41] on several levels: healthcare level (differences in the whole range of morbid physical and mental conditions associated with ACEs [42], in biological markers [43], in personal skills [44], and in the capacity for resilience [45]); economic level (an economic difference in terms of savings in health, school support and social expenditure [46], a difference in productivity [47], saving for the National Health System [48,49]); and social sphere (difference of outcomes related to daily functioning, work and need for social support [46,50]). The primary outcome is represented by the welfare-related health costs from birth to 3 years of life. The computerization of health care data in Lombardy, allows, through the tax code, to deduct health care services performed, and medications prescribed and globally deduct the subject’s health care costs. Costs will be tracked through the individual infant’s tax code until age 20, with successive steps: every 6 months until age 3, then 4, 5, 8, 11, 14, 17, and 20 years.

Secondary outcomes are welfare-related health costs from trauma detection to delivery and difference in the development of ACEs-related diseases and conditions.

During the follow-up it will be possible to detect the main ACEs-related diseases: children exposed to higher psychological stress have been shown to have higher risk of common diseases of childhood, including otitis media, viral infections, asthma, dermatitis, urticaria, intestinal infectious diseases, and urinary tract infections. Childhood adversities have also been associated with greater risk of adult chronic conditions, including cardiovascular disease, stroke, cancer (excluding skin cancer), asthma, chronic obstructive pulmonary disease, kidney disease, diabetes, overweight or obesity, and depression, as well as increased health risk behaviors [14,15,16].
Besides, it will be possible to describe difference concerning the number of children with secure attachment (at 18 months of age) [51]; differences in developmental scales (Bayley scale at 6, 18, 36 months) [52]; social outcome certification for handicap, need educator at school or home, disability certification, social-welfare benefits, skipped daycare days and sick days, how many workdays lost by parents due to parental care, number of accidents (home and non-home), prescription-drug fills, injury-insurance claims, and criminal convictions [53].

## 3. Conclusions

An overall rate of return on investment (ROI) statistically significant 13.0% per annum with an associated benefit/cost ratio (BCR) of 6.3 is expected as the primary outcome of the “WeCare Generation” program. These expectations are derived from the available scientific evidence [54,55,56].

ROI is one of the most widely used balance sheet indicators by investors and entrepreneurs to shed light on a company’s ability to deploy resources efficiently.

In other words, ROI measures the amount of money a company can generate after investing in any activity regardless of the type of funding sources used.

The benefit-cost ratio (BCR) is a profitability indicator used in cost-benefit analysis to determine the viability of cash flows generated from an asset or project. The BCR compares the present value of all benefits generated from a project/asset to the present value of all costs. A BCR exceeding one indicates that the asset/project is expected to generate incremental value.

The expected outcomes of the study are a statistically significant economic return and an improvement in the health of the treated population.

The economic results will be useful to increase the impact on welfare policy.

If European health systems will invest in the first thousand days of life, according to equity and effectiveness criteria, they will empower new generations, with a strong economic and health return.

## Figures and Tables

**Figure 1 ijerph-19-16741-f001:**
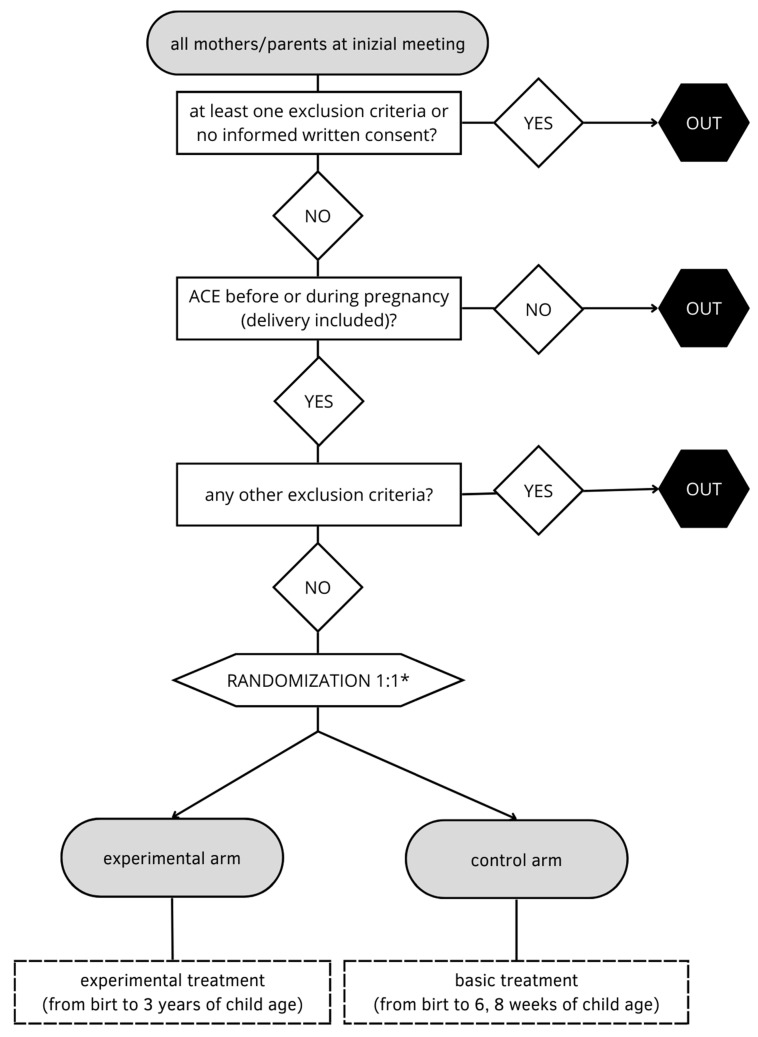
Study design. * Randomization will take place, as soon the ACEs will be detected.

## Data Availability

The data presented in this study will be available on request from the corresponding author.

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
