# Peer review of "1000 Days: The “WeCare Generation” Program—The Ultimate Model for Improving Human Mental Health and Economics: The Study Protocol"

_ijerph, 2022, doi:10.3390/ijerph192416741_

Round 1

Reviewer 1 Report

The work proposed with the links to ACEs links to the current discussion in several national context to childhood trauma and adverse experiences. However, the article submitted could provide more context and information so the reader can further understand the ACEs approach to wellbeing intervention.

There is need in the introduction to include evidence to support the bold statements made. There is no citations or evidence to support claims, for example.

In the past two years, the covid-19 pandemic has had a strong impact not only because of the effect of the virus on the human organism, but also as a global traumatic event (is there evidence?)  We know how trauma, for the individual, is a psychically unaddressable event that elicits and unconsciously 'repeats' past traumas (evidence?). A decisive factor, therefore, for individual resilience is how well past traumas, especially those involving childhood, have been processed. We also know how a reliable, caring environment during childhood is an element of resilience (evidence)

The We Care programme needs to be introduced further in the introduction to provide context.

In the Background section ACEs need to be explained and there could be further critique of the ACEs approach. Is this a globally recognized approach? At there any criticisms of the ACES approach? Are there specific adverse childhood experiences recognised as ACEs. ? Do different ACE impact each other?

The ethical considerations need to discussed in the methods and materials. 

The study materials and method are explained clearly. Useful to have the training discussed.

Tables need to be explained further.

Avoid emotive tone in places 'huge amount of resources' line 93

Providing more evidenced context will support the work.

Author Response

The work proposed with the links to ACEs links to the current discussion in several national context to childhood trauma and adverse experiences.

Point However, the article submitted could provide more context and information so the reader can further understand the ACEs approach to wellbeing intervention.

Response FIXED

Point There is need in the introduction to include evidence to support the bold statements made. There is no citations or evidence to support claims, for example.

In the past two years, the covid-19 pandemic has had a strong impact not only because of the effect of the virus on the human organism, but also as a global traumatic event (is there evidence?)  We know how trauma, for the individual, is a psychically unaddressable event that elicits and unconsciously 'repeats' past traumas (evidence?). A decisive factor, therefore, for individual resilience is how well past traumas, especially those involving childhood, have been processed. We also know how a reliable, caring environment during childhood is an element of resilience (evidence)

Response FIXED

Point  The We Care programme needs to be introduced further in the introduction to provide context.

Response FIXED

Point In the Background section ACEs need to be explained and there could be further critique of the ACEs approach. Is this a globally recognized approach? At there any criticisms of the ACES approach? Are there specific adverse childhood experiences recognised as ACEs. ? Do different ACE impact each other?

Response FIXED

PointThe ethical considerations need to discussed in the methods and materials. 

Response FIXED

Point Tables need to be explained further.

Response  since the tables were confusing, for clarity we removed the tables and left the contents in the text,-

Point Avoid emotive tone in places 'huge amount of resources' line 93

Response FIXED

Point  Providing more evidenced context will support the work

Response FIXED

Reviewer 2 Report

The subject of this paper seems very interesting, having the intention to combine two perspectives: health and economics. It is a good idea. Also, the epigenetics view is also important.

The paper refers to a study protocol, that means a research project instead of a research itself. The idea and the study aims and design can be interesting. Nevertheless, the manuscript has many weaknesses.

The abstract is so long and it is not in accordance with the journal format. It need to be shorten.

Keywords have acronyms, that is not appropriate and should be clarified.

The introduction makes a contextualization for the study, based on the pandemic situation and showing a holistic view and intention for a program to be developed. This section doesn’t present references. After follows a background section in which many references are indicated, but not well presented. It is difficult to found so many publications referring all to the same content. It is suggested to write a more detailed and specific background, indicating which studies refer precisely to specific issues. The sentences are vague and written as a common sense idea. For example “ACE are an important risk factor for various mental, physical and social (10,11,12,66,57,24,27,33)” is a consensual, known, but vague idea. More, it is need to specify which of the eight references are related to mental conditions, or which refers to physical, and soon. Furthermore, a logical sequence of references is required. These points should be applied to the whole text.

Material and Methods clarify the type of study and the centres to be involved. That’s ok. Inclusion and exclusion criteria are clear, as well as the recruitment procedures. The next items continue clear, the tools for data, etc. The phrases of lines 226-231 need theoretical support, since some declarations are made without theoretical background. Also EMDR therapy was not referred in the background and its selection to be applied on the trauma before birth and epigenetic perspective needs to be very well justified. Secondary outcomes, although its content, are confused in the text organization: the transition from line 301 to 302 is not well done, not understandable.

Tables 1 and 2 and are so long and not understandable. There is not description of its content and origin of data/results included.  

Conclusions are like reflexion and not based on the supposed results. Instead of being a study protocol, a good literature reviews can be done and presented, as well as results of the relation between economics and health, which references mentioned epigenetics, specific traumas, and soon.

So, a more careful text and specific protocol study is suggested. Authors also need to clarified how they guarantee the follow up until twenty years old and why they choose to investigate firstly in one-year period and after 5 years old choose to do it in three years’ periods. All this type of things need to be more organized and well structured. More important than the number of references is its use and adequacy. If major chances will be made, it can be a good protocol.

Author Response

Point  The abstract is so long and it is not in accordance with the journal format. It need to be shorten.

Response FIXED

Point  Keywords have acronyms, that is not appropriate and should be clarified.

Response FIXED

Point The introduction makes a contextualization for the study, based on the pandemic situation and showing a holistic view and intention for a program to be developed. This section doesn’t present references.

Response FIXED

 Point    After follows a background section in which many references are indicated, but not well presented. It is difficult to found so many publications referring all to the same content. It is suggested to write a more detailed and specific background, indicating which studies refer precisely to specific issues.

Response FIXED

 Point The sentences are vague and written as a common sense idea. For example “ACE are an important risk factor for various mental, physical and social (10,11,12,66,57,24,27,33)” is a consensual, known, but vague idea.

 More, it is need to specify which of the eight references are related to mental conditions, or which refers to physical, and soon. Furthermore, a logical sequence of references is required. These points should be applied to the whole text.

Response

the fact is that ACEs are related to so many different conditions within the physical, psychological and distributed pathology throughout the entire lifeline that it is not possible , dealing with the root of this complex tree , to make more focused statements.

On the other hand, we have tried to be precise by specifically naming each individual pathological or behavioral condition in the text.

 Point

Also EMDR therapy was not referred in the background and its selection to be applied on the trauma before birth and epigenetic perspective needs to be very well justified. Secondary outcomes, although its content, are confused in the text organization: the transition from line 301 to 302 is not well done, not understandable.

Response FIXED

 Point   Tables 1 and 2 and are so long and not understandable. There is not description of its content and origin of data/results included. 

Response  since the tables were confusing, for clarity we removed the tables and left the contents in the text,-

Point    Conclusions are like reflexion and not based on the supposed results. Instead of being a study protocol, a good literature reviews can be done and presented, as well as results of the relation between economics and health, which references mentioned epigenetics, specific traumas, and soon.

Response FIXED

Point 1: The work proposed with the links to ACEs links to the current discussion in several national context to childhood trauma and adverse experiences. However, the article submitted could provide more context and information so the reader can further understand the ACEs approach to wellbeing intervention.

Response 1: we introduced a background with respect to childhood adverse experiences

Point 2:  There is need in the introduction to include evidence to support the bold statements made. There is no citations or evidence to support claims, for example.

There is need in the introduction to include evidence to support the bold statements made. There is no citations or evidence to support claims, for example.

In the past two years, the covid-19 pandemic has had a strong impact not only because of the effect of the virus on the human organism, but also as a global traumatic event (is there evidence?)  We know how trauma, for the individual, is a psychically unaddressable event that elicits and unconsciously 'repeats' past traumas (evidence?). A decisive factor, therefore, for individual resilience is how well past traumas, especially those involving childhood, have been processed. We also know how a reliable, caring environment during childhood is an element of resilience (evidence)

Response 2:

Point 3:The We Care programme needs to be introduced further in the introduction to provide contex

Response 3:

Point 4; In the Background section ACEs need to be explained and there could be further critique of the ACEs approach. Is this a globally recognized approach? At there any criticisms of the ACES approach? Are there specific adverse childhood experiences recognised as ACEs. ? Do different ACE impact each other?

Response 4:

Point 5;The ethical considerations need to discussed in the methods and materials.

Response 5 :

Point 6: The study materials and method are explained clearly. Useful to have the training discussed

Round 2

Reviewer 1 Report

Some sections in literature review still are not given an evidence context or are too subjective / emotive.

As a global tragic event, the COVID-19 pandemic has posed a serious threat to global  mental health (any research to support). The current evidence and published literature related to previous epidemics suggest that mental health issues may arise after the peak of the pandemic (what research?), with increased prevalence among the vulnerable population and people with risk factors (which studies). Currently we observe an exponential increase in mental illnesses especially of the adolescent age, but also of the adult and early childhood ages (any research that highlights this?)  Growing demands for mental health, urgent social protection, and better public health systems have posed critical challenges in both economic and service delivery capacities (research evidence to support).

Therefore, on the basis of scientific evidence, the research team considered if they could develop an area-based preventive curative model, that leads to a significant decrease in the risk of developing major mental and physical disorders as well as simultaneously leading to economic savings? Furthermore, could a programme that leads to major resilience and productivity of the population also support inter-generations? 

The sections above should be revised. 

Author Response

As a global tragic event, the COVID-19 pandemic has posed a serious threat to global  mental health (any research to support). The current evidence and published literature related to previous epidemics suggest that mental health issues may arise after the peak of the pandemic (what research?), with increased prevalence among the vulnerable population and people with risk factors (which studies). Currently we observe an exponential increase in mental illnesses especially of the adolescent age, but also of the adult and early childhood ages (any research that highlights this?)  Growing demands for mental health, urgent social protection, and better public health systems have posed critical challenges in both economic and service delivery capacities (research evidence to support).

Requested references to support data have been added.

Therefore, on the basis of scientific evidence, the research team considered if they could develop an area-based preventive curative model, that leads to a significant decrease in the risk of developing major mental and physical disorders as well as simultaneously leading to economic savings? Furthermore, could a programme that leads to major resilience and productivity of the population also support inter-generations?

The sections above should be revised.

Sections have been revised. Thank you for your suggestions.

Reviewer 2 Report

The major suggestions was not taking in account. More references were added, but authors do not explore these references. It was only put in brackets. Ethical procedures are not appropriately described. How written consent will be registered? what is done for data collection? which ethical institution evaluate the study proposal?

For several suggestions the answer given was "Fixed". So, what does it mean, essentially when there were no changes? 

There are excessive number of references and the quality of the manuscript is not improved by this increase, but maybe by its interpretation. Moreover, the references are not well written, which shows also a low care with the text quality.

In this stage the paper doesn't reveal good conditions, instead of being a good idea for research. 

Author Response

The major suggestions was not taking in account. More references were added, but authors do not explore these references.

References have been reduced, corrected and discussed.

It was only put in brackets. Ethical procedures are not appropriately described.

How written consent will be registered? what is done for data collection? which ethical institution evaluate the study proposal?

Requested information has been added.

For several suggestions the answer given was "Fixed". So, what does it mean, essentially when there were no changes? 

Changes have been made.

There are excessive number of references and the quality of the manuscript is not improved by this increase, but maybe by its interpretation. Moreover, the references are not well written, which shows also a low care with the text quality.

In this stage the paper doesn't reveal good conditions, instead of being a good idea for research. 

Thank you for your useful suggestions. The paper has been revised and references have been corrected.